# Ultrafast ion sieving using nanoporous polymeric membranes

Pengfei Wang[1], Mao Wang[1], Feng Liu[1,2], Siyuan Ding[1], Xue Wang[1], Guanghua Du[3], Jie Liu[3], Pavel Apel[4], Patrick Kluth [5], Christina Trautmann[6] & Yugang Wang[1]

The great potential of nanoporous membranes for water filtration and chemical separation has been challenged by the trade-off between selectivity and permeability. Here we report on nanoporous polymer membranes with an excellent balance between selectivity and permeability of ions. Our membranes are fabricated by irradiating 2-μm-thick polyethylene terephthalate Lumirror® films with GeV heavy ions followed by ultraviolet exposure. These membranes show a high transport rate of $K^+$ ions of up to 14 mol h$^{-1}$ m$^{-2}$ and a selectivity of alkali metal ions over heavy metal ions of >500. Combining transport experiments and molecular dynamics simulations with a polymeric nanopore model, we demonstrate that the high permeability is attributable to the presence of nanopores with a radius of ~0.5 nm and a density of up to $5 \times 10^{10}$ cm$^{-2}$, and the selectivity is ascribed to the interaction between the partially dehydrated ions and the negatively charged nanopore wall.

[1] State Key Laboratory of Nuclear Physics and Technology, Peking University, 100871 Beijing, People's Republic of China. [2] Center for Quantitative Biology, Peking University, 100871 Beijing, People's Republic of China. [3] Institute of Modern Physics, Chinese Academy of Sciences, 730000 Lanzhou, People's Republic of China. [4] Flerov Laboratory of Nuclear Reactions, Joint Institute for Nuclear Research, Dubna, Russian Federation 141980. [5] Department of Electronic Materials Engineering, Research School of Physics and Engineering, Australian National University, Canberra, 2601, Australia. [6] GSI Helmholtzzentrum and Technische Universität Darmstadt, 64291 Darmstadt, Germany. P. Wang and M. Wang contributed equally to this work. Correspondence and requests for materials should be addressed to F.L. (email: liufeng-phy@pku.edu.cn) or to Y.W. (email: ygwang@pku.edu.cn)

Nanoporous membranes have attracted broad interest because of their great potential in many scientific areas and industry applications, including ion separation and water desalination[1–12]. Their performance is governed by two key factors: permeability and selectivity. However, the two properties are often inversely correlated[2,3]. To solve this problem, it has been a long-standing goal to fabricate membranes with a high density of nanopores with a uniform size approaching the molecular scale, i.e., ≤1 nm[3,4]. A variety of artificial membranes have been fabricated using both bottom–up and top–down approaches, but they are still outperformed in ion separation by their natural counterparts such as cell membranes with ion channels or pumps[13–16]. In these bio-membranes, the potassium ion channel KcsA shows a transport rate of $K^+$ ions up to $10^8$ ions $s^{-1}$ (corresponding to 6 mol $h^{-1}$ $m^{-2}$ assuming that the density of the membrane channel is $10^9$ $cm^{-2}$), and a $K^+/Na^+$ selectivity of >1000[17,18]. In contrast, bottom–up fabricated synthetic NF-270 membranes have a transport rate of $Na^+$ ions of up to 3.6 mol $h^{-1}$ $m^{-2}$ but exhibit nearly no $K^+/Na^+$ selectivity and a $K^+/Mg^{2+}$ selectivity of <2[19]. Lyotropic liquid crystal membranes show a $K^+/Mg^{2+}$ selectivity of ~33, but the transport rate of $K^+$ ions is only $1.3 \times 10^{-4}$ mol $h^{-1}$ $m^{-2}$[19]. The same is true for the membranes fabricated by a top–down approach. For example, the nanoporous monolayer graphene membrane fabricated by ion bombardment excels in high transport rates, yet is limited in ionic selectivity[15]. In a recent study, we have reported on polymer membranes demonstrating extremely high ionic selectivity, e.g., alkali metal ions over heavy metal ions of $10^4$. These membranes containing negatively charged pores with a radius of ~0.3 nm were fabricated with 12-μm-thick polyethylene terephthalate (PET) Hostaphan® films[20]. During fabrication, the membranes were irradiated with energetic ions. Each ion projectile produces a track, i.e., a narrow damage trail. Instead of using chemical etching of these tracks to form pores (as typically applied in the track-etching technique), sufficient extended ultraviolet (UV) exposure was applied (hence this fabrication method is named as the track-UV technique). Despite their excellent selectivity, the application of the PET Hostaphan® films seems to be limited because their transport rates of $K^+$ ions are only $2.0 \times 10^{-3}$ mol $h^{-1}$ $m^{-2}$. The compromise between permeability and selectivity also limits the performance of graphene oxide (GO) laminates, which have become promising membranes for ionic filtration[21]. As the interlayer spacing of GO is 1.3 nm, the transport rate of $K^+$ ions reaches 2.0 mol $h^{-1}$ $m^{-2}$ but the $K^+/Mg^{2+}$ selectivity is just 1[13]. On the other hand, when the interlayer spacing is reduced to <1.0 nm, the selectivity of $K^+/Mg^{2+}$ becomes comparable with or even better than that of the nanoporous Hostaphan® membranes, but the transport rate of $K^+$ ions drops to <0.007 mol $h^{-1}$ $m^{-2}$[14].

To improve the permeability and benefit from the high selectivity of ions, we tested much thinner films by using 2-μm-thick PET Lumirror® films. Compared to 12-μm-thick nanoporous Hostaphan® membranes, our nanoporous Lumirror® membranes show highly increased transport rates of alkali metal ions by >3 orders of magnitude to about 10 mol $h^{-1}$ $m^{-2}$ without significantly compromising the ionic selectivity. Based on molecular dynamics (MD) simulations and transport measurements, this improved performance is ascribed to several factors: the permeability is significantly enhanced mainly due to an increased pore radius to ~0.5 nm; the fine selectivity is maintained due to the electrostatic interaction between the charged pore walls and ions, coupled with the partial dehydration effect. Thus this artificial nanopore system marks a major advance to compete with its natural counterpart and shows great potential for industrial applications where ultrafast ionic sieves are required.

## Results

**High ionic transport rate and ionic selectivity.** We generated nanoporous PET Lumirror® membranes using our recently developed fabrication method, i.e., the track-UV technique[20]. Notably, after the irradiation of swift heavy ions with a fluence of $5 \times 10^{10}$ ions $cm^{-2}$ and 4-h UV exposure on both sides, the Lumirror® membranes maintain their flexibility and mechanical strength over a macroscopic area. Using our custom-built electrolytic cell (Supplementary Fig. 1a), we measured the transmembrane ion currents when applying a voltage across the film (Supplementary Fig. 1b) and confirmed that the electrolyte ion current increases as the UV treatment time increases (Supplementary Fig. 1c). Moreover, the current is proportional to the fluence of the ion irradiation (Supplementary Fig. 1d) and the concentration of electrolyte ions (Supplementary Fig. 1e).

By systematic measurements of the ionic transport rates of a variety of cations with inductively coupled plasma optical emission spectrometry (ICP), we identified dramatically increased ionic transport rates across the membranes compared with previous data reported for 12-μm-thick PET Hostaphan® membranes. Particularly, the transport rate of $K^+$ ions reaches an unprecedented high value of 14 mol $h^{-1}$ $m^{-2}$, outperforming all other reported membranes and surpassing the nanoporous PET Hostaphan® films by >6000 times. At the same time, the nanoporous PET Lumirror® membranes maintain high ionic selectivity. The cation transport rates clearly diverge into three groups (Fig. 1a). Alkali metal ions show the highest transport rates of >10 mol $h^{-1}$ $m^{-2}$ in the order of $Li^+ < Na^+ < K^+ < Rb^+(Cs^+)$; alkaline earth metal ions show medium transport rates between ~0.1 and ~0.45 mol $h^{-1}$ $m^{-2}$ in the order of $Mg^{2+} < Ca^{2+} < Ba^{2+}$; and heavy metal ions, like $Cu^{2+}$, $Fe^{2+}$, $Mn^{2+}$, $Cd^{2+}$ ions, show the lowest transport rate of ~0.01–0.03 mol $h^{-1}$ $m^{-2}$. The transport rates between the first two groups differ by about 1–2 orders of magnitude, whereas the difference is up to 3 orders of magnitude between the alkali metal ions and heavy metal ions. For ion separation, this is of interest, because heavy metal ions are considered as toxic at high concentrations[22]. The wide gap between different electrolyte ions demonstrates the high selectivity for ionic transport through the membrane (all tested chemicals are listed in Supplementary Table 1). Notably, even in the 1:1 binary $K^+/Mg^{2+}$ ionic mixture with a high total concentration of 1 M, the transport rate ratio of $K^+/Mg^{2+}$ is still about 20 (Supplementary Table 3).

The most remarkable result is that the nanoporous PET Lumirror® membranes exhibit both high ionic selectivity and high transport rates. Figure 1b compares the membrane of this study with the other reported nanoporous membranes regarding transport rates as well as selectivity of $K^+/Mg^{2+}$ or $Na^+/Mg^{2+}$[13–15,19,20,23–25]. To compare the performance of the membranes in applications such as ionic separation, the overall transport rates are shown instead of the normalized transport rate per channel. Although the density of the nanopores in the membrane (which is usually proportional to the overall transport rate) might vary between the different membranes, its maximum value reflects the upper limit allowed by the material strength and the fabrication method. Moreover, different driving forces (electric field, osmosis pressure, or air pressure) for ionic transport are applied to the different membranes. Since this prevents a comparison of the membranes under identical experimental conditions, the graph is limited to the best performance of each membrane, assuming it was measured under optimal performance conditions (e.g., highest air pressure that the membrane can endure). For all membrane data collected from the literature (Supplementary Table 2), the membrane with the next highest transport rate is the zwitterion-carbon nanotube membrane (ZCNT) followed by NF-270 or GO, their selectivity

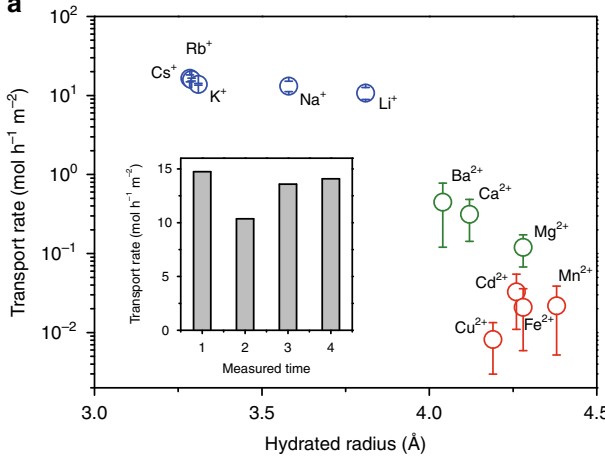

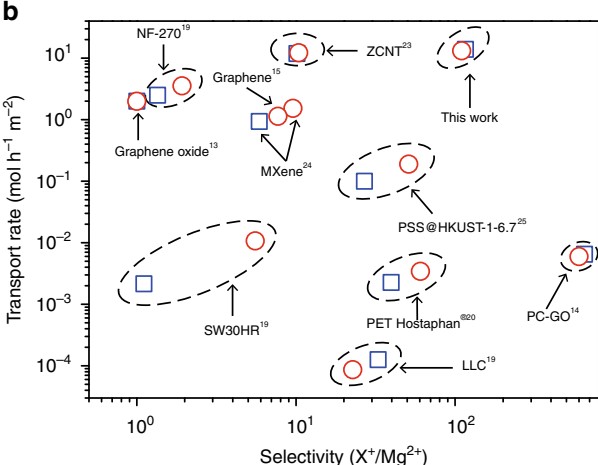

**Fig. 1** Transport rates and selectivity. **a** Transport rates of alkali metal ions (blue), alkaline earth metal ions (green), and transition metal and heavy metal ions (red) measured with ICP, as a function of the hydration radius of the ions. The transport rates are normalized to a 1 M feed solution and measured under 10 V applied voltage. Error bars represent the standard deviation of at least three independent measurements using different membrane samples. The inset shows the transport rates of $Na^+$ ions of four independent measurements. Unless otherwise specified, the membranes were 2-μm-thick PET Lumirror® membranes irradiated with 1.4-GeV Bi ions at a fluence of $5 \times 10^{10}$ ions $cm^{-2}$ and subsequently exposed to UV radiation for 4 h. **b** Transport rates vs. selectivity of various nanoporous membranes; the ionic sieving membrane studied in this study outperforms in both transport rates and ionic selectivity of $K^+/Mg^{2+}$ (blue) and $Na^+/Mg^{2+}$ (red)

being about 10 for ZCNT[23], and nearly no selectivity for the other two[13,19]. On the other hand, a recently reported physically confined GO membrane shows relatively small transport rates, but the selectivity of $K^+$ over $Mg^{2+}$ is the highest, nearly 1000[14]. Hence the nanoporous PET Lumirror® film simultaneously excels in both permeability and selectivity, demonstrating great application potential for ion separation.

**MD simulations using a polymeric nanopore model**. To understand the exceptional ionic sieving property of the membrane, we constructed an atomic model of a polymeric nanopore and tried to recapitulate the observed experimental results by MD simulations (Supplementary Fig. 2a). In contrast to simplified models such as carbon nanotubes used in transport simulations, our model pore captures the key features of the polymeric

nanopores. This includes the rough (instead of smooth) inner surface. As shown in Supplementary Fig. 2b, the inner radius varies from 0.4 to 0.55 nm. Moreover, discrete charges are randomly distributed (Fig. 2a). They originate from the deprotonated carboxyl groups formed during water immersion of the ion-irradiated and UV-exposed films[26].

By extensively exploring a variety of simulation conditions, we find that the relative order of the simulated transport rates is consistent with the measured experimental results (Fig. 2b) if we set the average pore radius to about 0.5 nm, the applied electric field to 0.3 V $nm^{-1}$, and the surface charge density to $-0.8$ e $nm^{-2}$. In the simulations, only ionic transport under steady state is taken into consideration as the simulated flux for various ions increases linearly with time (Supplementary Fig. 2c).

The transport experiments indicate that the nanopores in the Lumirror® membranes are negatively charged, consistent with previously reported negatively charged nanopores[20,27,28]. First, our nanoporous membranes show a selectivity of cations over anions (Supplementary Table 1), which is also confirmed in our MD simulations. Moreover, we observed a strong pH dependence of the measured transport rates. The transport rate quickly drops from 1 mol $h^{-1}$ $m^{-2}$ for pH >3.5 by nearly two orders of magnitude for pH <1.5. The curve of the pH-dependent transport rate is very similar to the titration curve observed in 12-μm-thick Hostaphan® films except that the curve is shifted toward lower pH values. Consistently, the simulated ion flux of $K^+$ ions drops by nearly a factor of 10 as the surface charge density changes from $-0.8$ e $nm^{-2}$ to 0 e $nm^{-2}$ (Fig. 2c).

We note that the ions are transported together with water molecules. The ratio of the number of the water molecules to the $K^+$ ions inside the nanochannel is ~32 (Supplementary Fig. 2d), but the ratio of the transport number is only about 6 (Fig. 2d). To validate the transport model in our simulation, we inserted a capillary in our conductivity cell and monitored the height of water in the column (see Supplementary Fig. 1g and more details in Supplementary Note 1). In agreement with our simulations, these dedicated measurements confirmed the $H_2O/$ $K^+$ ratio of about 6, a value similar to the total number of the hydration water molecules for $K^+$ ions[29].

**Transport mechanism**. In order to explore the ionic transport mechanism and better understand the origin of the observed high ionic transport selectivity and ultra-high permeability of nanoporous Lumirror® membranes, we performed transport experiments with various ions and molecules, as well as the corresponding MD simulations.

The size of the nanopore is one of the key factors in determining the transport mechanism. Although the nanopore size is usually determined by imaging with transmission electron microscope or scanning electron microscope, it has been very challenging to achieve sufficient spatial resolution and contrast to image these nanopores in 2-μm-thick PET Lumirror® film. Nevertheless, several lines of evidence suggest that they have a larger radius than in 12-μm-thick PET Hostaphan® films. First, the order of the transport rate of the alkali metal ions switches from $Cs^+ < K^+ < Na^+ < Li^+$ for Hostaphan® films to $Li^+ < Na^+ < K^+ < Cs^+$ for Lumirror® films. Second, to correctly model the key transport property in the MD simulations, the radius of the polymeric nanopores in the Lumirror® film is required to be about 0.5 nm. Third, this result is corroborated by transport rates of organic salts of different sizes. The transport rate of the methylene blue ion is only $3.8 \times 10^{-4}$ mol $h^{-1}$ $m^{-2}$ and its effective hydration radius is around 0.7 nm or even smaller if its asymmetric shape is taken into consideration (Supplementary Fig. 3). In contrast, tetrabutylammonium ions ($Bu_4N^+$) has a hydrated radius of only 0.49 nm but a transport rate of

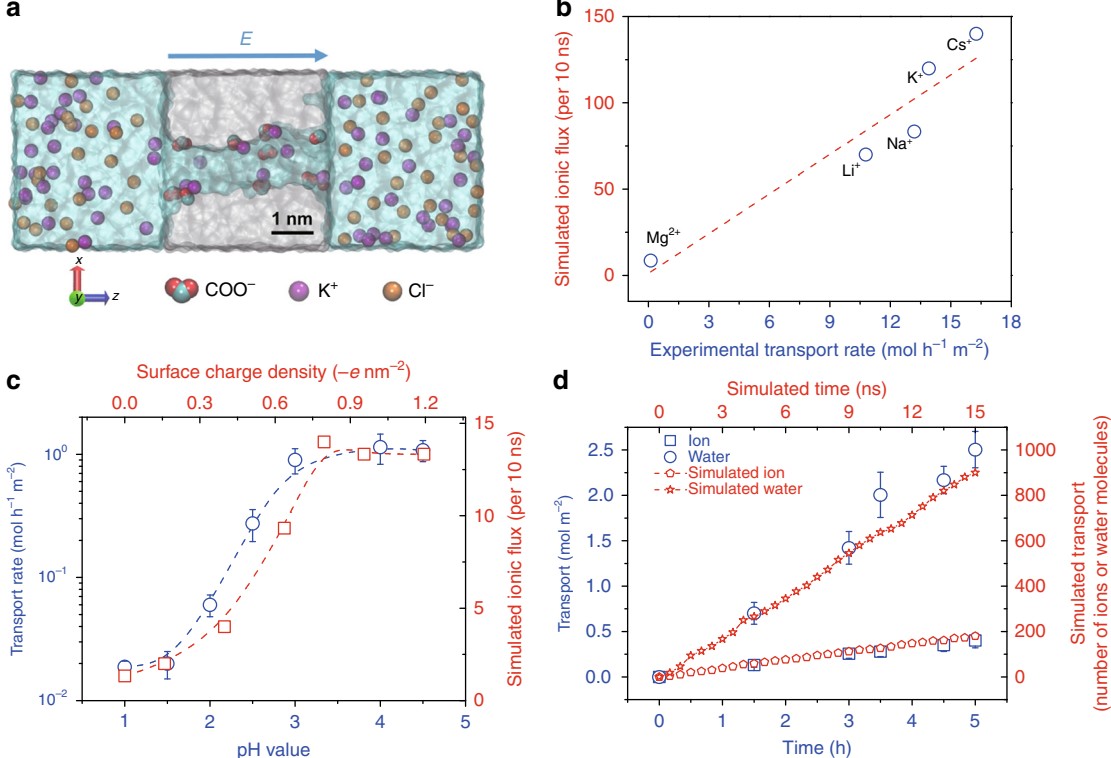

**Fig. 2** MD simulations and transport measurements. **a** Cross-sectional scheme of polymeric nanopore model with 1 M KCl electrolyte solution on both membrane sides. The polymer matrix material is shown in gray; negatively charged residues (COO⁻) are shown in red (O) and cyan (C); K⁺ and Cl⁻ ions are shown in purple and yellow, respectively. For all simulations without specification, a voltage of 1.2 V is applied along the pore axis, i.e., $z$ direction. **b** Simulated vs. experimental transport rates for various ions. The red dashed line is a linear fit to the data (blue circles) ($R^2 = 0.88$). **c** Experimental (circles) and simulated (squares) transport rates of K⁺ ions as a function of pH (blue) and surface charge density (red), respectively. Error bars represent the standard deviation of three measurements. The average transport rate of K⁺ ions through the PET Lumirror® films was measured with a feed solution of KCl (1 M) and an applied voltage of 5 V. A voltage of 0.12 V is applied across the pore in the simulation. The blue and red dashed lines are fits to the experimental and simulation data, respectively. **d** Comparison of simulated and experimental transport number of ions or water molecules. Water and ion transport through the PET Lumirror® film from the feed chamber with a 0.01 M KCl solution and an applied voltage of 10 V in the experiment. Dashed lines are linear fits

0.3 mol h⁻¹ m⁻², almost three orders of magnitude larger than the rate of methylene blue ions. Furthermore, the transport rate of the tetraethylammonium ion (Et₄N⁺) with a radius of 0.4 nm increases by another factor of 5. Notably, the bare radii of these organic cations are much larger than the radii of metal ions, resulting in a negligible hydration shell (their hydration radii are nearly the same as the bare ion radii) (Supplementary Table 1). Regarding transport rates of these organic cations, size exclusion seems to play a more dominant role than charge or dehydration effects. Hence these organic ions serve as a suitable "ruler" to probe the size of our nanopores. We note that the transport rate of Et₄N⁺ through the nanopores with a radius of about 0.3 nm in 12-μm-thick PET Hostaphan® films is below the detection limit.

The size of the nanopore has a direct influence on selectivity and permeability of ionic transport[14,30–33]. As illustrated in Fig. 3a, on the one hand, when the pore size is smaller than the hydration radius of the ions, dehydration becomes dominant as shown in nanoporous PET Hostaphan® films or in physically confined GO membranes with subnanometer interlayer spacing[14,20]. Such nanopores often show high selectivity but low permeability of ions (left panel, Fig. 3a). On the other hand, if the pore size is much larger than the Debye screening length (~1–100 nm) of the electric double layer, the transport rate is determined by the bulk mobility of the ions. These nanopores often show high permeability of ions but nearly no selectivity between cations and anions or between different cations (right panel, Fig. 3a).

Neither scenario of the ionic transport discussed above can account for the nanopores in our PET Lumirror® membranes. Since the radius of the nanopores in the PET Lumirror® membranes is about 0.5 nm, which is larger than the hydration radii of most of the metal ions tested (Supplementary Table 1), it is unlikely that a significant dehydration barrier exists due to steric hindrance at the entrance of the nanopores. Actually, based on MD simulations the first hydration shell of the transported K⁺ ions is nearly intact (Fig. 3e and Supplementary Fig. 2e). On the other hand, the nanopores in the PET Lumirror® films are not large enough to form a complete electrical double layer to screen the surface charges on the walls for the ions to transport with their bulk mobility. In fact, the negatively charged nanopores show a selectivity of cations over anions (Supplementary Table 1) and the observed transport rates are not correlated with the bulk mobility of the ions.

The balance between high permeability and high selectivity of ions observed on the nanoporous PET Lumirror® films could result from the transport scenario demonstrated in the middle panel of Fig. 3a. On the one hand, compared to 12-μm-thick PET Hostaphan® films, the permeability of K⁺ ions through the PET Lumirror® films improves by 1000 times because the dehydration barrier is significantly lowered due to increased pore size[33–35]. Also, the shorter length of the pores helps to increase permeability[36,37]. On the other hand, the observed high selectivity between the alkali metal ions and alkali earth metal ions or heavy

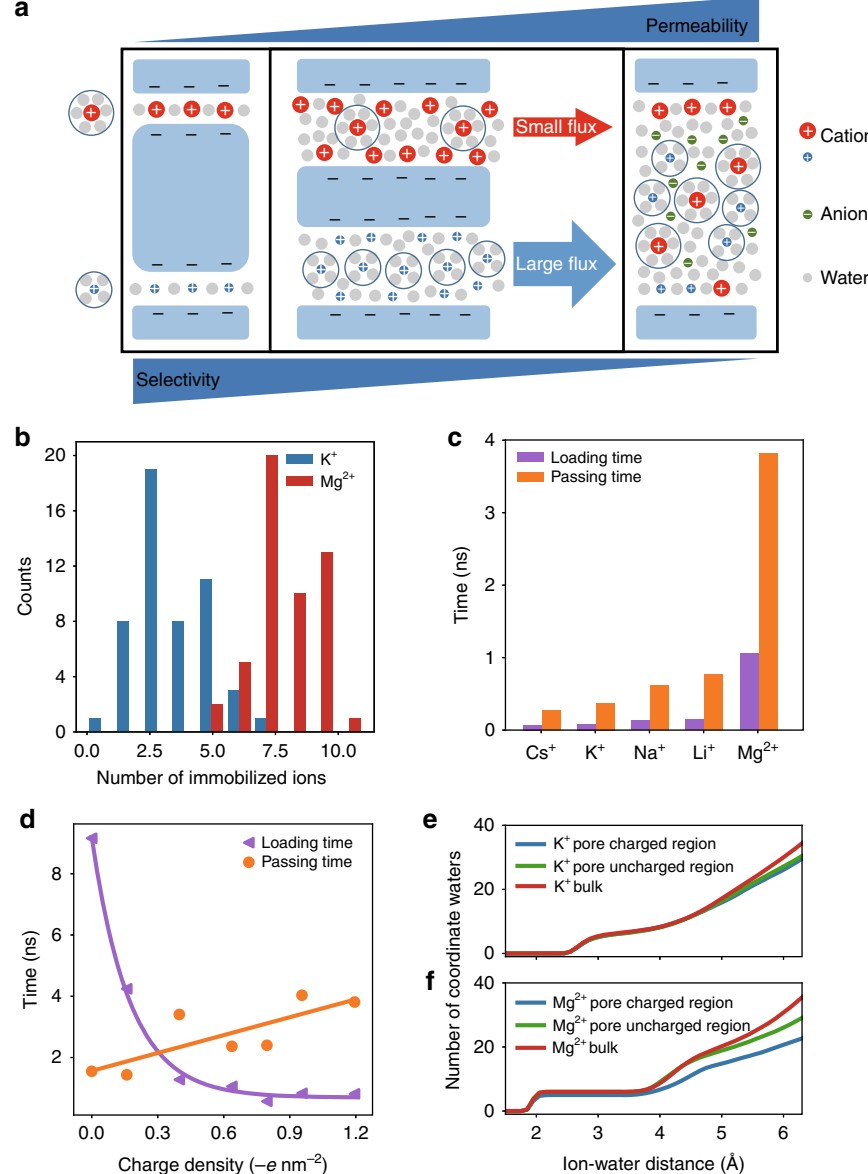

**Fig. 3** Simulated ionic transport phenomena. **a** Schematic of three representative ionic transport mechanisms as the pore size increases in nanopores with negatively charged sidewalls (blue). When the pore size is less than the hydration radius of the cations, the dehydration energy determines the ionic selectivity. The nanopore shows great selectivity but low permeability (left panel). As the pore size is much larger than the Debye screening length, the nanopore shows great permeability but no ionic selectivity (right panel). When the pore size is between the two cases, the nanopore shows high transport rate of one cation and high selectivity between the two cations if their interaction with the pore surface is different (middle panel). This represents the scenario of the transport mechanism of the nanoporous Lumirror® films. **b** Histogram of immobilized $K^+$ (blue) and $Mg^{2+}$ (red) ions on the inner surface of the nanometer pore. **c** Average loading time (purple) and passing time (orange) for $Cs^+$, $K^+$, $Na^+$, $Li^+$, and $Mg^{2+}$ ions. **d** Average loading time (purple) and passing time (orange) of $K^+$ ions as a function of surface charge density of the nanopores. Dashed lines are the respective linear and exponential fits of the passing and loading times. **e**, **f** Cumulative radial distribution function of water molecules surrounding $K^+$ ions (**e**) and $Mg^{2+}$ (**f**) in the bulk solution (red), inside the nanopore close to the uncharged region (green) without negative inner surface charges or charged region (blue) with negative inner surface charges and trapped ions

metal ions could be contributed significantly by the following two factors: one is the electrostatic interaction between the ions and the negatively charged pore wall, and the other is the partial dehydration of the transported ions. The MD simulations reveal two different behaviors: some ions get trapped close to the charges on the inner surface, while other ions pass swiftly through the channel (Supplementary Movies 1 and 2). The number of immobilized $Mg^{2+}$ ions is twice as large as that of $K^+$ ions (Fig. 3b). This is consistent with the observation that the passing time of the transported $K^+$ ions through the nanopore is much

shorter than that of $Mg^{2+}$ ions (Supplementary Fig. 4a). Both the average passing time and the average loading time (time elapsed between two ions sequentially entering the nanopore) is significantly larger for $Mg^{2+}$ ions than for alkali metal ions (Fig. 3c). Therefore, the transport rate of $K^+$ ions is much higher than that of $Mg^{2+}$ ions. Moreover, as the surface charge density changes from $-1.2$ e $nm^{-2}$ to $0$ e $nm^{-2}$, the passing time of $K^+$ ions gradually decreases by a factor of 2, whereas the loading time increases exponentially (Fig. 3d). This is probably the reason why the transport rate of $K^+$ ions drops at low pH (Fig. 2c). To further

confirm the role of the surface charge in ion selectivity, we performed ionic transport experiments as the pH value of the solution was tuned from high to low till the negative surface charges on the nanopore wall were neutralized. We find that the selectivity of $K^+/Mg^{2+}$ ions remains nearly unchanged but drops when the pH value is <4 (Supplementary Fig. 1f). On the other hand, the smaller transport rate of $Mg^{2+}$ ions may also be influenced by partial dehydration inside the nanochannel. MD simulations show that some of the hydration water molecules are stripped from the outer hydration shells, and occasionally even the first hydration shells of the transported $Mg^{2+}$ ions (Fig. 3f and Supplementary Fig. 2f), indicating that $Mg^{2+}$ ions are partially dehydrated. This process is even more pronounced when the ions pass nearby the charged region with inner surface charges and transiently trapped ions, which narrow the nanochannel. Compared to alkali metal ions, alkali earth and heavy metal ions experience stronger adsorption and their hydration radii are larger. During transport, they are thus more frequently and more severely dehydrated resulting in a lower transport rate.

## Discussion

To fabricate membranes with both high permeability and high ion selectivity, it is important to control the size of the nanopores and provide a large number of uniformly sized nanopores. This requirement has been a long-standing challenge in the fabrication of nanoporous membranes[3]. For example, phase inversion membranes such as microfiltration and ultrafiltration have polydisperse pore size distributions[38]. In contrast, track-etched membranes have a very uniform pore size[39,40] but are limited in low pore density[41] ($<10^9$ cm$^{-2}$) due to the stochastic pore distribution and the risk of overlapping of neighboring pores[42]. In addition, the pore sizes of these membranes are too large for ionic separation. Interestingly, PET Lumirror® membranes with unetched but UV-treated tracks contain the nanopores with a radius of nearly 0.5 nm and a density of up to $5 \times 10^{10}$ cm$^{-2}$. And they show highly reproducible ionic transport measurement results, suggesting that these nanopores are nearly uniform in size. The transport rates of all tested ions are measured at least three times using different samples. For all alkali metal ions, the relative standard deviations are <20% (Supplementary Fig. 5a and Supplementary Table 1), which is a rather small variability for transport measurements through nanopores. Excluding systematic uncertainties from the experiment, the variability of the transport rate is probably even smaller.

The uniform size of the nanopores is an intrinsic property given by highly controllable fabrication process using the track-UV technique (Fig. 4a). Compared with the conventional track-etching technique, this fabrication process consists of the irradiation of high-energy heavy ions, and subsequent extended UV exposure, but no chemical etching. The high-energy heavy ions (such as Bi, Au, or U ions) of a fixed kinetic energy in the GeV range pass through many micrometer-thick polymer films nearly un-deflected. Each incident ion deposits uniformly energy along its trajectory resulting in an individual track (Supplementary Fig. 5b), which consists of the highly damaged material (amorphization, broken bonds, and reduced mass density due to outgassing degradation fragments). These parallel oriented tracks have a lateral diameter of several nanometers[43] and often a highly damaged small core[20]. Moreover, the threshold of energy loss for the formation of homogeneous and continuous cylindrical tracks in polymer membranes is between 2 and 5 keV nm$^{-1}$[44]. The fact that we irradiated our membranes with 1.4 GeV Bi ions further helps to generate the small distribution of pore size. According to SRIM-2010 calculations[45], the mean energy loss of the ions is about 16 keV nm$^{-1}$, which is well above the threshold yielding

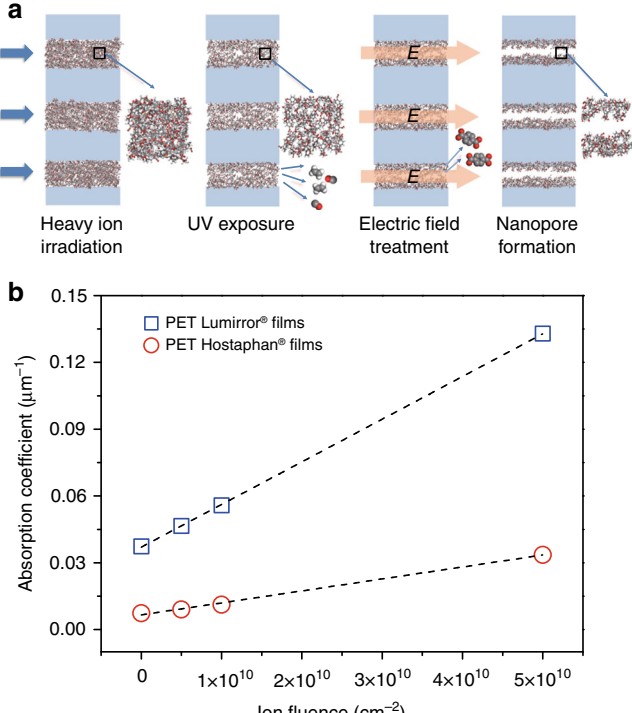

**Fig. 4** Fabrication process of nanoporous membranes with the track-UV technique. **a** Schematic of the nanopore formation process in a thin slice of the polymer film irradiated with energetic heavy ions and subsequent UV radiation. Each ion projectile produces a nanometric cylindrical track (in atomic representation) inside the polymer film (light blue). Given the collimated ion beam (blue arrows), the tracks are oriented parallel and stochastically distributed. Tracks are amorphous cylinders and consist of broken polymer chains and decreased density due to the release of small volatile degradation molecules such as $C_2H_4$ and CO during ion irradiation and UV-induced photochemical reactions[56]. The charged fragments such as terephthalic acids are further cleaned when an electric field is applied across the membrane immersed in water. Finally, uniform nanopores form inside the core of the track. **b** UV absorption (at 365 nm) as a function of ion fluence of PET Lumirror® (blue square) and Hostaphan® (red circle) films. Dashed lines are linear fits

homogeneous damage trails. Furthermore, the energy deposition within the thin film varies by <2% and the energy of the incident ion beam is fixed within 0.5%.

However, even with the homogeneous and continuous cylindrical tracks, it is still very difficult to generate a high density of uniformly sized pores at the nanometer scale with chemical etching. As mentioned above, chemical etching of tracks leads to pores with the size too large for the ion separation. Instead, we modified ion tracks by sufficiently long UV exposure. UV-induced effects are ascribed to photo-destruction of ion-beam-induced radiolysis products resulting in an additional release of small volatile molecules and accumulation of small fragments such as terephthalic acids that are extracted in aqueous environment[46]. When our membranes are immersed in the electrolyte and a voltage is applied, small fragments are extracted from the tracks and thus increase their free volume. We found that sufficient UV exposure is the key to generating the nanopores in both PET films Hostaphan®[20] and Lumirror®. The transport rates increase sharply to a high level as the UV treatment time increases (Supplementary Fig. 1c and ref. [20]).

The UV absorption coefficient of Lumirror® films (Supplementary Fig. 6) is directly correlated with the ion fluence

(Fig. 4b). This effect is known for many polymers and is ascribed to carbonization of the track material leading to a decrease of the bandgap[47]. Interestingly, the absorption coefficients of both pristine and irradiated Lumirror® films are stronger than that of Hostaphan® films (Supplementary Table 4). The reason is unclear, but we speculate that small quantities of additives such as the antioxidant may play a role. It is well known that special additives are often added into the polymers to improve their quality. In fact, PET Lumirror® films have the highest tensile strength among all plastic films and can be fabricated to films as thin as 2 μm. They also show high radiation resistance revealing no apparent loss of structural integrity after 4-h UV treatment. In contrast, PET Hostaphan® films usually become too fragile if the UV treatment time is >3 h.

From the current practical point of view, the nanoporous PET Lumirror® film still have room for improvement. Although, compared with other selective membranes, they show a higher transport rate of $Li^+$ ions, their selectivity of $Li^+/Mg^{2+}$ is only 90, which is smaller than a recently reported metal-organic framework membrane[25] and nanoporous PET Hostaphan films[20]. This limits the application of the nanoporous PET Lumirror® film in the extraction of $Li^+$ ions from salt-lake brines. However, nanoporous polymer films are still attractive due to their robustness, flexibility, and efficiency in large-scale production. Particularly, large-scale production is essential for real applications but has been challenging for most of the nanoporous membranes fabricated in bench-top-type experiments[48]. To date, there exist several large-scale accelerator facilities where commercial irradiation of large amounts of ion-tracked polymer films (e.g., $2 \times 10^6 m^2$ per year) takes place. The newly developed track-UV fabrication method outperforms the conventional fabrication method for generating a high density of uniform nanopores with the radius approaching the molecular size. We expect that, using other polymer films or composite materials, it is possible to tune the selectivity and permeability and finally reach the demand of large-scale real-world applications such as extraction of $Li^+$ ions or the potential to compete with their natural counterparts.

In conclusion, by irradiating 2-μm-thick PET Lumirror® films with high-energy heavy ions followed by extended exposure to UV radiation, we fabricated nanoporous membranes of high permeability and ion selectivity outperforming previously reported systems. The new nanoporous membrane achieves an excellent balance between permeability and selectivity as it exploits both the interactions between the ions and the pores as well as the partial dehydration effect. These observations provide new insights into the transport mechanisms as the pore size approaches the molecular scale (Fig. 3a). The mechanisms discovered in this unexplored regime for the studies of ion transport will deepen our understanding of nanofluidics. In addition, the films have great structural strength, flexibility, and can be fabricated efficiently using well-established mass production processes. They have great potential to be applied as ultrafast ion sieves for ion separation and water filtration.

## Methods

**Fabrication of nanopores in Lumirror® membranes.** Polymer films of PET (Lumirror, C60B, Toray Industries, 2 μm thick) and PET Hostaphan® films (Hostaphan, RN12, Hoechst, 12 μm thick) were irradiated under vacuum at room temperature with 1.4-GeV Bi ions at the Heavy Ion Research Facility of HIRFL (Lanzhou, China). The irradiation fluence ranged from $5 \times 10^9$ to $5 \times 10^{10}$ ions $cm^{-2}$. The track density on the membrane irradiated at low fluence was measured with the density of the pores made of the track-etching technique, and it was confirmed to be consistent with the irradiation fluence. Each side of the irradiated films was exposed to UV radiation (MUA-165, Meijiro Genossen, Japan) with a wavelength ranging from 280 to 600 nm (peaked at 365 nm) and with a flux density of 4.2 mW $cm^{-2}$. Unless otherwise stated, the exposure time per film side was 4 h. Before transport measurements, UV-exposed membranes were kept in air for at least 1 week.

**Electrical and transport measurements.** The ion transport experiments were conducted with a custom-made conductivity cell (Supplementary Fig. 1a). Nanoporous PET films were placed between the two 1.5 mL chambers of the cell. A bias of several volts was applied across two platinum electrodes. For the measurements of conductance currents, the two chambers were filled with 0.1 M electrolyte solutions and the electric currents were recorded with a Keithley electrometer (6517B, Keithley, USA). For the measurements of ionic transport rates, the feed and permeate chambers were filled with 1 M salt solution and deionized water, respectively. ICP–atomic emission spectroscopy (Profile Spec, Leeman, USA) for cations and IC (ICS-90, Dionex, USA) for anions were applied to analyze the remaining solution in the permeate chamber to determine the ionic transport rate. For pH measurements, the pH value of electrolyte solutions was titrated with HCl from 5 to about 1. No ion conductance and ion transport rate change could be detected after the pH value was adjusted back to 5. The membrane condition remained the same before and after ion transport experiments. The pH of the solution did not change over the whole process of ion transport measurements. Since the total ion transport was much smaller than the feed amount, the concentration change during the measurement was negligible. The transport rates of the organic ions were measured in the same way, except that the solution concentration was 0.1 M, which is the saturated concentration of a methylene blue solution.

**UV absorption and UV-visible spectrum measurements.** For pristine and irradiated PET Hostaphan® and Lumirror® films, absorption measurements were performed in the wavelength range of 200–800 nm using a UV-visible/near infrared Spectrophotometer (Hitachi UH4150). The absorption coefficients were calculated with the Beer–Lambert law after correction for the reflectivity (see the Supplementary Note 2 for more details).

**MD simulation.** The atomic representation of the polymeric nanopore (Supplementary Fig. 2a) was adapted from a model designed previously[49]. In brief, the polymer bulk was built using a collapsing–annealing method with several PET chains (each chain included 9 subunits). A polymer cell was simulated in a constant number of particles, pressure, and temperature (NPT) ensemble for 500 ps at a pressure of 1 atm and a temperature of 1000 K. Then the temperature was reduced to 400 K with a cooling step of 100 K every 200 ps. After that, the polymer was equilibrated at 300 K for 500 ps. The final density of the polymer was 1.15 g $cm^{-3}$, which is close to the experimental value of 1.40 g $cm^{-3}$. Atoms were removed to form an approximately 1 nm wide and 4 nm long nanopore. Negatively charged residues ($COO^-$) were randomly distributed on the inside surface of the nanopore. Then the polymeric membrane was solvated with electrolytes. The size of the whole simulation system was $4.8 \times 4.2 \times 11.5 nm^3$. Ions were added by replacing randomly selected water molecules to keep the electro-neutrality of the whole simulation system and match with the 1 M concentration in the bulk solution.

All MD simulations were performed using GROMACS4.6[50] with CHARMM36[51] force field. The TIP3P water model[52] was used. The important interaction parameters for all the simulated metal ions are listed in Supplementary Table 5. And our simulation package is available upon asking. Electronic continuum correction was added to the lithium ions to account for the electronic polarizability of water in ionic solutions[53]. The simulation used 1 fs time steps and Particle-Mesh Edward electrostatics. Van der Waals interactions were calculated using a cutoff of 1 nm. The temperature was maintained at 300 K using v-rescale. Simulations were first equilibrated for 200 ps in NPT ensemble at a pressure of 1 atm with the membrane constrained. The last frame of the MD equilibration was used as the starting structure for further simulations with the constant number of particles, volume, and temperature (NVT) ensemble. Periodic boundary conditions were applied to all three directions.

For NVT simulations, a voltage bias was applied along the z direction (track axis). To save computing time, we used a relatively strong electric field of 0.3 V $nm^{-1}$ to speed up the transport processes (the order of the transport rate between different ions remains unchanged as we decrease the electric field) and fixed the membrane to prevent it from moving. To simulate the pH dependence of the nanopore, six different systems were prepared with different carboxyl numbers on the inner surface of the nanopore. The carboxyl numbers are 0, 2, 5, 8, 10, and 15, corresponding to the densities of negative surface charges of 0, 0.16, 0.40, 0.64, 0.80, 0.95, and 1.20 -e $nm^{-2}$, respectively. We used the later 15 ns of the total 20 ns simulation for data analysis. To reduce the effect of a strong electric field, we lowered the electric field to 0.03 V $nm^{-1}$ in the pH dependence simulation.

To count the immobilized ions inside the nanopore, we compared the displacement of the ions in two subsequent frames and analyzed the expected displacement r, which is defined by the diffusion formula $r = \sqrt{6Dt}$, where D is the diffusion coefficient of a cation and t is the time interval[54]. We chose t = 0.3 ns and assumed the value of D is 1.957 and 0.705 $nm^2 ns^{-1}$ for $K^+$ and $Mg^{2+}$, respectively[55]. During the time interval, cations with a displacement below r were counted as immobilized cations.

**Data availability.** All data generated or analyzed during this study are included in this published article and its supplementary information files.

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

## Acknowledgements

The authors thank Qi Ouyang, Yiqing Gao and Wei Guo for valuable discussions and the staff at the Heavy Ion Research Facility of HIRFL (Lanzhou, China) for preparation and irradiation of PET Lumirror® films. This work was supported by National Science Foundation of China (Grant Nos. 11335003 and 31670852) and the National Magnetic Confinement Fusion Energy Research Project of China (2015GB113000). F.L. acknowledges the support from Peking University's 100-talent plan. P.K. acknowledges the Australian Research Council for financial support. Christina Trautmann acknowledges support from the Deutsche Forschungsgemeinschaft (DFG-FOR1583). The MD simulation was performed on the High Performance Computing Platform of the Center for Life Sciences, Peking University.

## Author contributions

F.L. and Y.W. conceived the research. P.W., S.D., and X.W. performed the experiment and data analysis. M.W. performed MD simulations. G.D. and J.L. assisted in sample

irradiation. F.L., Y.W., C.T., P.A., P.K., P.W., and M.W. prepared the manuscript. All authors discussed the results, commented on the manuscript, and contributed to the writing of the paper.

## Additional information

**Competing interests:** The authors declare no competing financial interests.

