## [Peer Review File · Nature Communications]

Reviewers' comments:

Reviewer #1 (Remarks to the Author):

In this study, the authors reported a nanoporous polymeric membrane which was synthesised by heavy ion irradiation and UV exposure. The membrane exhibited high selectivity on certain ion. MD was also conducted to understand transport of various ions in the nanopores. The membrane synthesis methods is interesting. The discussion part is quite weak and it is very hard to follow. The paper could be published after some improvements on experiment, simulation and discussion to address the following questions/concerns.

1. In introduction, the authors misunderstand the trade-off of current membrane in water processing (water treatment, seawater desalination). In water processing, the permeability refers to the transport rate of water molecule, rather than ions. The ultimate goal is to let water molecule to transport faster and reject all ions and other contaminants.
2. In Figure 2.d, the unit of simulated flux is missing.
3. As shown in Figure 2.c, pH has significant impact on the transport rate of ions. The authors need to conduct additional experiment to investigate the impact of pH on the selectivity;
4. On page, the authors claimed that size exclusion play larger role than dehydration and charge in determining the transport rate. It is quite easy to understand the role of nanopore size in transport rate. The authors may need to conduct some new experiment to understand the roles of dehydration and surface charge in the transport rate using membranes with same size pore, but different other properties.
5. On the page 11, the authors concluded that the interaction between ions and the negatively charged internal surface of nanopore contribute significantly to the selectivity. What kind of interactions are involved between ions and internal surface?
6. What is the concentration of tested salt in feed chamber?

Reviewer #2 (Remarks to the Author):

The authors prepared nanoporous 2 μm thick polyethylene terephthalate (PET) Lumirror® films by a track-UV etched process as reported by the authors previously (12 μm PET Hostaphan® film) Adv. Funct. Mater. 2016, 26, 5796). The resulted membrane shows high transport rate of alkali ions and high selectivity for $\text{K}^+/\text{Cu}^{2+}$ under electric field, due to the different size of the hydrate ions and negatively charged pore surface. The performance is improved by just replacing the 12 μm PET Hostaphan® film by 2 μm PET Lumirror® films. The reasons for the improvement are not clearly demonstrated. Therefore, the novelty and significant are degraded.

Comments:

1. For determine the pore size, it is better to do BET measurement, or explore small neutral molecules.
2. Are there no free water is the channels since the author claimed that the amount of water to K^+ is six as that of hydrated K^+ ions? This is very strange. If the hydrated ions passed one by one without free water, how fast could it be?
3. For the separation performance, the separation performance of the binary ionic mixture should provide.
4. What are the additives in the PET Lumirror® films, which can induce so significant difference from PET Hostaphan® film?
5. For the comparison, it is not only list $\text{Na}^+/\text{Mg}^{2+}$, $\text{K}^+/\text{Mg}^{2+}$ or $\text{K}^+/\text{Cu}^{2+}$, actually, $\text{Li}^+/\text{Mg}^{2+}$ should provide. By the way, emphasize the $\text{K}^+/\text{Cu}^{2+}$ up to 1000 is no meaning. Therefore, please remove such part from the abstract, introduction and conclusion.
6. The concentration and electric field used for simulation is 1 M and 0.3 V/nm, respectively, while which are 0.1 M and 0.03 V for the transportation test, why?
7. Some citation are not consistent as they cited in the references list.

8. Why (in)Fig2d, the simulation flux without units and the time scale are different from the experiment?
9. What are the Debye lengths of the corresponding ions?
10. It is not good to emphasize the desalination in the manuscript, since the present membrane has lack potential desalination. Therefore, the corresponding parts need to be revised.
11. The K^+ transport rate is not as high as $20 \text{ Mol h}^{-1} \text{ m}^{-2}$ as stated by the authors. It need clarify.
12. Why the transport rate of Cu^{2+} is less than Cd^{2+} ?
13. For practical application, it is better to perform the prepared membrane for ionic separation through pressure driven process. It might be more attractive.
14. To investigate the transportation mechanism, additional the electrochemical impedance plots are requested

Reviewer #3 (Remarks to the Author):

Review of "Ultrafast sieving of ions using nanoporous polymeric membranes" submitted to Nature Communications.

This manuscript presents an approach to synthesize a nanoporous membrane by subjecting a polyethylene terephthalate (PET) film to swift heavy ion irradiation followed by ultraviolet (UV) light exposure. The experimental studies are complemented by molecular dynamics simulation of ion transport through a nanoporous polymer. The work is innovative and demonstrates that nanoporous polymers can be synthesized with good permeability and Na^+/Mg^{2+} ion selectivity. The work will be of interest to the readers of Nature Communications. However, there are several issues with the manuscript.

1. The industrial application is hyped in page 3. It is not clear how swift heavy ion irradiation with GeV ion beams is scalable or how one can produce inexpensive membranes by this process. The key advance here is the control over pore size. Recently, Abraham et al [15] have demonstrated such control with physically confined graphene oxide (PCGO).
2. The abstract claims that this membrane has potential to address the global challenges of water scarcity. To achieve that, the membrane must separate ions from water. This membrane does not do that. It lets alkali metal ions to pass through along with water (One ion for every six water molecules). Abraham et al [ref. 15] have recently synthesized a membrane that dramatically reduces ion transport (including Na^+), especially when the layer spacing drops below 0.74 nm.
3. The comparison made in the introduction between cell membranes and synthetic membranes is muddled. The authors use the unit ion/s for the former and $\text{mol}/(\text{hm}^2)$ for the latter, which makes the comparison difficult. In addition, the authors mention K^+/Na^+ selectivity for the former and K^+/Mg^{2+} separation for the latter.
4. The idea of obtaining improved sieving when a layer spacing or pore diameter is reduced below 1 nm is not new. Several authors, including ref. 14 and 15, have explored this concept previously.
5. Abraham et al [15] and Devanathan [in the same issue of Nature Nanotechnology] have discussed the issue of hydrated ions, hydration shell diameter and hydration energy to explain the selectivity observed in previous studies. A version of the schematic representation of ion transport with different pore sizes, presented in Figure 3a, has been previously presented by Abraham et al and Devanathan.
6. Figure 1b is misleading because it does not include the recent PCGO result [15], which shows a K^+/Mg^{2+} and Na^+/Mg^{2+} selectivity of almost 1000.
7. Within the alkali metal ion family, there is no selectivity. All of these ions pass through the membrane easily. Since the authors mention K^+/Na^+ selectivity in the introduction, this point cannot be ignored.
8. On page 12, the authors suggest that "the nanopores in the PET Lumirror film in this study are nearly uniformly sized". There is no direct evidence to back up this claim from material

characterization. How uniform are the pores? What is the spacing between pores?

9. The statement on page 15 that the additives in PET Lumirror are responsible for enhanced UV absorption is highly speculative. What are these additives?

10. In Table S1, the hydrated radius of methylene blue ion should be 7x3 Angstrom instead of 0.7x0.3 Angstrom.

11. The authors should provide details of the chemical structure of the polymer and force fields used in the Supplementary Information. This will help other researchers reproduce the work.

Overall, this paper may be considered for publication in Nature Communications after required revision. This work has to be placed properly in the context of previous work, especially recent findings in ref. 14 and 15.

Summary of Specific Changes Made to the Manuscript in Response to the Reviews

To make it as easy as possible for the Editor, we pasted the reviews verbatim below. Our response to each criticism/suggestion is depicted in green font and the exact changes to the manuscript sections are pasted in quotations as they are found in the revised manuscript, mostly in red font. The locations of these modifications in the revised manuscript are highlighted as yellow.

Reviewer #1 (Remarks to the Author):

In this study, the authors reported a nanoporous polymeric membrane which was synthesised by heavy ion irradiation and UV exposure. The membrane exhibited high selectivity on certain ion. MD was also conducted to understand transport of various ions in the nanopores. The membrane synthesis methods is interesting. The discussion part is quite weak and it is very hard to follow. The paper could be published after some improvements on experiment, simulation and discussion to address the following questions/concerns.

1. In introduction, the authors misunderstand the trade-off of current membrane in water processing (water treatment, seawater desalination). In water processing, the permeability refers to the transport rate of water molecule, rather than ions. The ultimate goal is to let water molecule to transport faster and reject all ions and other contaminants.

Thank you very much for your comment. We are sorry for the confusion. Although our new membrane excels at removing the toxic heavy metal ions from water, it still does not fit the ultimate requirement for water desalination. Hence in our paper the permeability does not refer to the transport rate of water molecules, but the selected ions. To avoid the potential confusion, we removed the word “water desalination” from the abstract, changed the last sentence in the abstract from “Thus the nanoporous PET Lumirror[®] membrane has great potential to be applied as an ultrafast ionic sieve to address the global challenges of water scarcity” to “Thus these UV-treated nanoporous ion-track membranes have great potential for applications where ultrafast ionic sieves are required”. We added “in ion separation” in the sentence of “A variety of artificial membranes have been fabricated using both bottom-up and top-down approaches, but they are still outperformed in ion separation by their natural counterparts such as cell membranes with ion channels or pumps” in the introduction (paragraph 1 on page 2).

2. In Figure 2.d, the unit of simulated flux is missing.

Thank you very much for your careful reading. We corrected the label of Figure 2d to be “Simulated transport (number of ions or water molecules)”.

3. As shown in Figure 2.c, pH has significant impact on the transport rate of ions. The authors need to conduct additional experiment to investigate the impact of pH on the selectivity.

Thank you very much for your insightful suggestion. We measured the ionic transport at low pH and found the selectivity drops. For example, the selectivity of K^+/Mg^{2+} decreases from more than 100 at $pH \geq 4$ to about 5 at pH 2. As the transport rate of K^+ ions decreases significantly, the transport rate of Mg^{2+} ions remains nearly unchanged. These results confirm that pH has great impact on the selectivity, indicating the electrostatic interaction plays an important role in the transport mechanism. We added **Supplementary Fig. 1f** to show the ionic selectivity as a function of pH, and rewrote the part in discussing the selectivity mechanism **on page 9-12**, in particular, we added a few sentences in **the first paragraph on page 12** “To further confirm the role of the surface charge in ion selectivity, we performed ionic transport experiments as the pH value of the solution was tuned from high to low till the negative surface charges inside the nanopores were neutralized. We find that the selectivity of K^+/Mg^{2+} ions remains nearly unchanged but drops when the pH value is below 4 (Supplementary Fig. 1f).”

4. On page, the authors claimed that size exclusion play larger role than dehydration and charge in determining the transport rate. It is quite easy to understand the role of nanopore size in transport rate. The authors may need to conduct some new experiment to understand the roles of dehydration and surface charge in the transport rate using membranes with same size pore, but different other properties.

Thank you very much for your insightful suggestion. To investigate the effect of the surface charge, we conducted transport experiments with the same nanopores at low pH when the negative surface charges are neutralized. The results indicate that the electrostatic interaction between the ions and surface charges plays an important role in ionic selectivity. On the other hand, since the size of the nanochannel is only slightly larger than the hydration radius of the ions, the partial dehydration could also contribute to ion selectivity. Based on our MD simulations, we analyzed the hydration of the transported ions in the uncharged region without inner surface charges, and the charged region with the negative inner surface charges and the trapped ions. For the latter, we needed to increase the temporal sampling rate to extract the distribution of the hydration water. The new results show that although the first hydration shell of the transported K^+ ions is nearly intact, some of the hydration water molecules are stripped from the outer hydration shells, and occasionally even the first hydration shells of the transported Mg^{2+} ions. Compared to alkali metal ions, alkali earth and heavy metal ions could experience stronger adsorption and their hydration radii are larger. During transport they are thus more frequently and more severely dehydrated resulting in a lower transport rate. Taken together, we think both the size and the surface charge of the nanopores are important in determining ion transport rate, and electrostatic interaction and dehydration effect both contribute to the selectivity. On the other hand, the size exclusion plays a dominant role in the transport of **organic ions** through the nanopores. Because organic ions have much weaker electrostatic interaction with the nanopore wall, they are less likely trapped inside

the nanopore. And they have very thin hydration shells, hence the dehydration effect could be negligible.

To clarify this issue, we replaced the original Figure 3e to the cumulative radial distribution function of the hydration water molecules surrounding K^+ or Mg^{2+} ions to show the partial dehydration effect, and added the corresponding radial distribution plots as Supplementary Fig. 2e-f. We rewrote the first paragraph on page 12: “The smaller transport rate of Mg^{2+} ions may also be influenced by partial dehydration inside the nanochannel. ... During transport they are thus more frequently and more severely dehydrated resulting in a lower transport rate.”

5. On the page 11, the authors concluded that the interaction between ions and the negatively charged internal surface of nanopore contribute significantly to the selectivity. What kind of interactions are involved between ions and internal surface?

Since the size of the negatively charged nanopore is only slightly larger than the size of hydrated ions, the interactions involved between ions and internal surface of the nanopore are a little bit complicated. Based on our MD simulations, due to the electrostatic interaction between the transported ions and the surface charges, some of the ions get trapped near the charge on the wall. These immobilized ions also contribute to the electrostatic interactions between the transported ions and the nanopore. Furthermore, our MD simulations suggest that the interaction between these trapped ions and the transported ions especially for the larger heavy metal ions could result in partial dehydration. Hence we conclude that both the electrostatic interaction and partial dehydration effect contribute to the selectivity. To clarify this issue, we revised the manuscript accordingly as we showed in the answer to the last remark.

6. What is the concentration of tested salt in feed chamber?

We are sorry for the confusion. The concentration of tested salts in feed chamber is 1 M for the ionic transport measurement except for methylene blue ions, which is saturated at 0.1 M. We added this information in the Material and Method section (paragraph 2 on page 17).

Reviewer #2 (Remarks to the Author):

The authors prepared nanoporous 2 μm thick polyethylene terephthalate (PET) Lumirror® films by a track-UV etched process as reported by the authors previously (12 μm PET Hostaphan® film) Adv. Funct. Mater. 2016, 26, 5796). The resulted membrane shows high transport rate of alkali ions and high selectivity for $\text{K}^+/\text{Cu}^{2+}$ under electric field, due to the different size of the hydrate ions and negatively charged pore surface. The performance is improved by just replacing the 12 μm PET Hostaphan® film by 2 μm PET Lumirror® films. The reasons for the improvement are not clearly demonstrated. Therefore, the novelty and significant are degraded.

Comments:

1. For determine the pore size, it is better to do BET measurement, or explore small neutral molecules.

Thank you very much for your insightful suggestions. We tried the BET measurements on the PET Lumirror® film with a Micromeritics ASAP 2020. Our preliminary results indicate that the specific surface area of the PET Lumirror® film is $37 \text{ m}^2/\text{g}$, much less than the other reported porous materials (usually more than $1000 \text{ m}^2/\text{g}$, for example, Walton, K.S. and Snurr, R.Q., JACS 129,8552-8556 (2007); Furukawa, H., et al., Science, 341,1230444-5 (2013)). And we are very surprised to observe a type III absorption isotherm with the Lumirror® film. This result seems to suggest that no pores have been detected on the Lumirror® film except for the nanometer scale surface craters, which clearly contradicts to the transport experiments with ions and the neutral molecules in solution. We speculate that this might be related with the relative small specific surface area of the PET Lumirror® film, or the special condition (e.g., in the solution under an electric field) for the opening of the nanopores in this film. To clarify this strange result, we need to run more experiments. Unfortunately we submitted nearly all the samples left from the last irradiation experiment for the BET measurement. To clarify the pore distribution with BET, we may need to wait for the next round of irradiation experiment, which will be scheduled 6 month later, to prepare sufficient amount of samples to try again.

Following your excellent suggestion, we also performed ion conductance measurements by adding small neutral molecules, e.g., series of polyethylene glycol (PEG) with the molecular weight from 62 to 300 g/mol. We observed the increase of the current by 50% if the radius of these neutral molecules is less than 0.45 nm, but no change when the radius of PEG molecules is larger than 0.55 nm. As a control, for PEG molecules with the radius from 0.25 nm to 0.6 nm, we observed the current increment for the membranes with nanopores of 15 nm in radius (see the figure below). These results strongly indicate that the nanopore is about 0.5 nm in radius in the 2 μm thick PET Lumirror® film. We think the current increment might result from the reorganization of the hydration bonds (Tunuguntla, R. H. et al., Science 357, 792–796 (2017)) inside the nanopore as the PEG molecules diffuse into the nanopore, contradicting to the thought that the PEG molecules inside the nanopore block the current (Krasilnikov, O. V., et al. Phys. Rev. Lett. 97,

018301 (2006)). In combination with MD simulations, we are still trying to collect more samples for test to validate this hypothesis.

(a-b) Ion current change of I_{PEG}/I as a function of the radius of the added PEG molecules when PET Lumirror® films (a) or PC films (b) were tested. I and I_{PEG} are the balanced ion current through the tested films before and after adding PEG solutions, respectively. Error bars represent the standard deviation of three measurements. 1 M KCl solution and deionized water were injected into the feed and permeate chamber respectively. A voltage of 1 V was applied across the film. The current jumped to a high peak as a drop of 0.1 ml PEG solution was added to the feed chamber when the current reached equilibrium. The PET Lumirror® films used in the measurements were irradiated with 1.4 GeV Bi ions at a fluence of 1×10^{10} ions cm^{-2} and subsequently exposed to UV radiation for 4 hours. The PC films used here were GVS Poretics™ Polycarbonate Track Etched membranes with nanopores with the radius of 15 nm at the pore density of 1×10^6 cm^{-2} .

(c,e) Ion current trace through PET Lumirror[®] films before and after adding PEG 200 (c) or PEG 300 (e) as a function of time. The blue line is the extension of the balanced ion current value after adding PEG solution for eye guiding.

(d,f) Ion current trace through PC films before and after adding PEG 200 (d) or PEG 300 (f) as a function of time.

2. Are there no free water in the channels since the author claimed that the amount of water to K⁺ is six as that of hydrated K⁺ ions? This is very strange. If the hydrated ions passed one by one without free water, how fast could it be?

Based on the MD simulations, although on average 6 water molecules are transported out together with one K⁺ ion, there are on average 194 “free” water molecules and 6 K⁺ ions inside the nanopore. The figure below shows the water molecules inside the nanopore. To clarify this concern, we added Supplementary Fig. 2d to show the number of water molecules and ions inside the nanopore during simulation, and we also added a sentence “The ratio of the number of the water molecules to the K⁺ ions inside the nanochannel is ~32 (Supplementary Fig. 2d)” in paragraph 3 on page 8.

Snapshot of the water molecules inside the nanopore in the MD simulation in the cross-section. Water molecules are shown in red (O) and white (H); K⁺ and Cl⁻ ions are shown in purple and orange, respectively. The polymer matrix material is shown in cyan.

3. For the separation performance, the separation performance of the binary ionic mixture should provide.

Thank you very much for your suggestions. We used the ICP to perform the separation measurement on the binary ionic mixtures, which are more close to the solutions in real applications. We found that the separation selectivity of K⁺/Mg²⁺ drops to 20. This is a common phenomenon for most of the other membranes (for example, Guo, Y., et al., *Angewandte Chemie International Edition* 55, 15120–15124(2016)). We added a sentence “Notably, even in the 1:1 binary K⁺/Mg²⁺ ionic mixture with a high total

concentration of 1 M, the transport rate ratio of K^+ / Mg^{2+} is still about 20 (Supplementary Table 3).” in the first paragraph on page 4.

4. What are the additives in the PET Lumirror® films, which can induce so significant difference from PET Hostaphan® film?

We have contacted the company making the film, Toray Industries, Inc., regarding the additives causing the difference between Lumirror® and Hostaphan® films, but they replied that they “cannot disclose Lumirror's production information in detail because it is confidential.” Based on our experience, we think the most important additives added in Lumirror® films could be some antioxidants such as Irganox. These antioxidants are usual strong UV absorber, and are efficient scavengers of radicals, including atomic hydrogen. Hence it could prevent hydrogen radicals from reacting with unsaturated carbon functional group. This explains why Lumirror® films have higher UV absorption coefficient before and after irradiation and it shows high resistance to the UV irradiation. We have tried to use FTIR to figure out the origin of the difference between the two films but no extra peaks were identified, suggesting the amount of the additives might be too small to be detected by IR. We did try to extract the additives from the films (Marie-Clarire Hennion, Journal of Chromatography A, 856,3-54 (1999)) and use Ultra-performance liquid chromatography-electrospray ionization mass spectrometry (UPLC-ESIMS) to identify the specific molecules (see the figure below). Since we do not have enough (at least 1 g) Hostaphan® films for this measurement, we chose a PET film similar to the Hostaphan® film as the control sample. It has the same thickness as the PET Hostaphan® film but is made by a different company. We observed multiple peaks with strong UV absorption, which might be the additives. We confirmed that Lumirror® films have much stronger UV absorption than the other PET films. And the relative composition of the additives appears to be different in the two films. We could identify a new peak (marked by the blue arrow) at the retention time of 3.45 min and two much stronger peaks (marked by the red circle) at 3.56-3.62 min. Furthermore, we found that the compounds associated with these peaks absorb the UV light at 239 ± 20 nm and have the molecular weight of about 300-400. But we still need more dedicated experiments to specify the molecules. To clarify this issue, we rewrote paragraph 2 on page 15.

Chromatograms of PET Lumirror[®] films (upper) and a 12-µm-thick PET films (bottom) analyzed with UPLC-ESIMS equipped with an ACQUITY H-Class UPLC (Waters Corp.) and a quadrupole rods SQ Detector 2 mass spectrometer (Waters Corp.). Separation were realized by a BEH C18 column (ACQUITY UPLC, 1.7 µm, 2.1×50 mm) using ultrapure water and acetonitrile as the mobile phase.

5. For the comparison, it is not only list Na⁺/Mg²⁺, K⁺/Mg²⁺ or K⁺/Cu²⁺, actually, Li⁺/Mg²⁺ should provide. By the way, emphasize the K⁺/Cu²⁺ up to 1000 is no meaning. Therefore, please remove such part from the abstract, introduction and conclusion.

Thank you very much for your suggestions. We totally agree with you that the selectivity ratio of Li⁺/Mg²⁺ is very important for the extraction of Li⁺ ions from salt-lake brines. We added our results and all the other available results on Supplementary Table 2. We also added a paragraph on page 15-16 to show the comparison “From the current practical point of view,This limits the application of the nanoporous PET Lumirror[®] film in the extraction of Li⁺ ions from salt-lake brines”. In addition, we replaced the selectivity ratio of K⁺/Cu²⁺ with a selectivity ratio between the alkali metal ions and heavy metal ions. The latter is often considered as toxic metal ions if the concentration is too high (Fu, F. & Wang, Q., Journal of Environmental Management 92, 407–418 (2011)).

6. The concentration and electric field used for simulation is 1 M and 0.3 V/nm, respectively, while which are 0.1 M and 0.03 V for the transportation test, why?

The application of higher concentrations and electric field in the simulation could help to speed up the transport processes hence shorten the total simulation time. This is a widely used in the MD simulations (Sahu, S., et al. Nano Lett. 17, 4719–4724 (2017); Luan, B., et al. Phys. Rev. Lett. 104, 238103 (2010); Sigalov, G., et al, Nano Lett. 8, 56–63 (2008)). To avoid the potential artifacts associate with the increased electric field (Köpfer, D. A, et al. Science 346, 352–355 (2014)), we confirmed that the order of the transport rate between different ions remain unchanged as we decrease the electric field. We added a note in the Material and Methods section (paragraph 2 on page 19).

7. Some citation are not consistent as they cited in the references list.

Thank you very much for careful reading. We have corrected all the citations and updated the reference list.

8. Why (in)Fig2d, the simulation flux without units and the time scale are different from the experiment?

We added the units of simulated flux in the updated Figure 2d. As we explained in question 6, we speed up the ion transport in the simulation compared with the experiment.

9. What are the Debye lengths of the corresponding ions?

Thank you very much for your insightful question. The Debye length is usually ~1-100 nm under most actual ionic conditions. But for 1 M KCl and MgCl₂ solution, the calculated Debye length is ~0.3 nm and ~0.17 nm, respectively, and these values could be even smaller if we consider the potential difference between the relative static permittivity in water and in nanopores. Since the radius of the nanopores in the PET Lumirror[®] film is only 0.5 nm, which is almost comparable with the size of the ions, it is unlikely that a complete double electrical layer could form inside the nanopores. Considering this is important for discussing the transport mechanism, we added several sentences “On the other hand, the nanopores in the PET Lumirror[®] films are not large enough to form a complete electrical double layer. . . . and the observed transport rates are not correlated with the bulk mobility of the ions. ” in first paragraph on page 11.

10. It is not good to emphasize the desalination in the manuscript, since the present membrane has lack potential desalination. Therefore, the corresponding parts need to be revised.

Thank you very much for your suggestions. We agree with you that the selection property of the current membrane is better suitable for the ion separation applications. It could be useful in removing the toxic heavy metal ions from the water, but not alkali metal ions. Hence to avoid the possible confusion, we revised the paper to be more focused on the ion separations by removing the relevant parts as we showed in the answer to the first remark of Reviewer 1.

11. The K^+ transport rate is not as high as $20 \text{ Mol h}^{-1} \text{ m}^{-2}$ as stated by the authors. It needs clarifications.

We corrected it to be the mean value of $14 \text{ mol h}^{-1} \text{ m}^{-2}$. Accordingly, the low limit of the selectivity of alkali metal ions over heavy metal ions is revised to 500, not 1000 (Abstract, page 1).

12. Why the transport rate of Cu^{2+} is less than Cd^{2+} ?

We are still trying to understand why Cu^{2+} ions happen to have the smallest transport rate in all the ions we tested. Although the hydrated radius of Cu^{2+} ions is slightly smaller than Cd^{2+} ions, the hydration energy of Cu^{2+} ions is $-2010 \text{ kJ mol}^{-1}$, the largest among all the ions tested. Since the hydrated radii of both ions are close to the pore size and they could be severally absorbed by the negative charges on the nanopore wall, it is very likely both ions are partially dehydrated during transport and the dehydration barrier of Cu^{2+} ions happens to be larger. However, we could not run a MD simulation to directly test this hypothesis because the transport rates are too low and the accurate parameters describing Cu^{2+} and Cd^{2+} ions are unavailable. Nevertheless, we follow your suggestion in question 5 and removed the selectivity of K^+/Cu^{2+} in the revision.

13. For practical application, it is better to perform the prepared membrane for ionic separation through pressure driven process. It might be more attractive.

Thank you very much for your insightful suggestions. We totally agree with you that the pressure driven separation is much more attractive considering its higher energy efficiency. However, such a measurement requires a significant upgrade of the current permeation measurement apparatus. This is beyond the scope of the current paper and will be addressed in our future work.

14. To investigate the transportation mechanism, additional the electrochemical impedance plots are requested.

Thank you very much for your suggestion. We performed the electrochemical impedance measurement using an Autolab PGSTAT302N potentiostat/galvanostat. The results and experimental details are shown below. The Nyquist and Bode plots show that the electrochemical impedance is different for the Lumirror[®] films in the solution of KCl , MgCl_2 and CrCl_2 . Interestingly, the magnitude of the electrochemical impedance at high frequency is positively correlated with the transport rate of the different electrolytes. However, since we do not have much experience, we have great difficulty to build a reasonable equivalent circuit model to account for all the observations. And we are not sure how to connect the parameters of the impedance element with the ionic transportation through the polymer nanopores with the radius of 0.5 nm (considering the usually applied continuum theory such as Nernst-Planck and Poisson equations may breakdown). We feel that it would be better to reveal the transportation mechanism with

more thorough systemic work in the future.

Nyquist (upper) and Bode (bottom) plots for the PET Lumirror[®] membranes in 0.1 M KCl (black), MgCl₂ (red) and CdCl₂ (blue). The films used in the measurements were irradiated with 1.4 GeV Bi ions at a fluence of 1×10^{10} ions cm⁻² and subsequently exposed to UV radiation for 4 hours. EIS data were collected with an Autolab PGSTAT302N potentiostat/galvanostat. The electrochemical cell in a three-electrode configuration contains a Ag/AgCl reference electrode, a Pt counter electrode and a Pt working electrode. All measurements were carried out with a bias voltage of 1 V on the basis of the cell equilibrium voltage at 25 °C in our conductance measurement apparatus. A sine-wave signal perturbation with the amplitude of 10 mV was applied in the 10 mHz–100 kHz frequency range.

Reviewer #3 (Remarks to the Author):

This manuscript presents an approach to synthesize a nanoporous membrane by subjecting a polyethylene terephthalate (PET) film to swift heavy ion irradiation followed by ultraviolet (UV) light exposure. The experimental studies are complemented by molecular dynamics simulation of ion transport through a nanoporous polymer. The work is innovative and demonstrates that nanoporous polymers can be synthesized with good permeability and Na⁺/Mg²⁺ ion selectivity. The work will be of interest to the readers of Nature Communications. However, there are several issues with the manuscript.

1. The industrial application is hyped in page 3. It is not clear how swift heavy ion irradiation with GeV ion beams is scalable or how one can produce inexpensive membranes by this process. The key advance here is the control over pore size. Recently, Abraham et al [15] have demonstrated such control with physically confined graphene oxide (PCGO).

Thank you very much for your comment. To clarify the concern on the scalable production of the irradiated polymer films, we added the relevant reference and a sentence “To date, there exist several large-scale accelerator facilities where commercial irradiation of large amounts of ion tracked polymer films (e.g., 2×10^6 m² per year) takes place.” in paragraph 1 on page 16. The figure below shows the irradiation system, which is capable to produce 8000 m² tracked polymer films (pore density: 10⁵ cm⁻²) in one hour with the same accelerator irradiating our PET samples at the Heavy Ion Research Facility of HIRFL. On the other hand, we agree with you that the key advance of this work is the development of a reliable fabrication method to control the pore size on the nanometer scale in polymer membranes. We believe that by further improving our technique and the method demonstrated in PCGO, we might be able to add new types of membranes for applications in ion separation and water desalination.

System for irradiating polymer films at the Heavy Ion Research Facility of HIRFL (Lanzhou, China)

2. The abstract claims that this membrane has potential to address the global challenges of water scarcity. To achieve that, the membrane must separate ions from water. This membrane does not do that. It lets alkali metal ions to pass through along with water

(One ion for every six water molecules). Abraham et al [ref. 15] have recently synthesized a membrane that dramatically reduces ion transport (including Na⁺), especially when the layer spacing drops below 0.74 nm.

Thank you very much for your comment. We agree with you that PCGO is better in water desalination than our new PET membrane although the new membrane is good at removing the toxic heavy metal ions from the water. Hence in the introduction and rest of the paper, we emphasize the selectivity and permeability of different ions. To avoid the potential confusion, we revised the paper as we stated in the answer to the first remark of Reviewer 1.

3. The comparison made in the introduction between cell membranes and synthetic membranes is muddled. The authors use the unit ion/s for the former and mol/(hm²) for the latter, which makes the comparison difficult. In addition, the authors mention K⁺/Na⁺ selectivity for the former and K⁺/Mg²⁺ separation for the latter.

Thank you very much for your comment. Assuming the density of the membrane channel is 10⁹ cm⁻², the transport rate of ions is 6 mol h⁻¹ m⁻² for cell membranes. We added this number in the introduction for a direct comparison (paragraph 1 on page 2). Since most of the synthetic membranes have no K⁺/Na⁺ selectivity, we chose the K⁺/Mg²⁺ selectivity as a common comparison standard for the synthetic membranes. To avoid the confusion, we added “exhibit nearly no K⁺/Na⁺ selectivity” in paragraph 1 on page 2.

4. The idea of obtaining improved sieving when a layer spacing or pore diameter is reduced below 1 nm is not new. Several authors, including ref. 14 and 15, have explored this concept previously.

We agree with you that we can improve selectivity by reducing the pore diameter or layer spacing below 1 nm. Just like the membranes you mentioned here, our previous nanoporous Hostaphan[®] films also show exceptional selectivity. However, the major concerns for these membranes are their low permeability. This is why we think this new Lumirror[®] film may have a distinct advantage by keeping an excellent balance between the selectivity and permeability.

5. Abraham et al [15] and Devanathan [in the same issue of Nature Nanotechnology] have discussed the issue of hydrated ions, hydration shell diameter and hydration energy to explain the selectivity observed in previous studies. A version of the schematic representation of ion transport with different pore sizes, presented in Figure 3a, has been previously presented by Abraham et al and Devanathan.

Thank you very much for your comment and the recommendation of a very important reference, **which was added as a citation in the revision**. We notice the similarity between the Figure 3a and the figure in Devanathan’s paper. But our figure focuses on the different transport mechanism between different ions, whereas Devanathan’s figure shows the mechanism of perfect desalination.

6. Figure 1b is misleading because it does not include the recent PCGO result [15], which shows a K⁺/Mg²⁺ and Na⁺/Mg²⁺ selectivity of almost 1000.

Thank you very much for your suggestion. Sorry for the confusion. We added the data point of PCGO in revised Figure 1b and change the sentence to be “a recently reported physically confined graphene oxide (PCGO) membrane shows relatively small transport rates, but the selectivity of K⁺ over Mg²⁺ is the highest, nearly 1000” in paragraph 1 on page 6.

7. Within the alkali metal ion family, there is no selectivity. All of these ions pass through the membrane easily. Since the authors mention K⁺/Na⁺ selectivity in the introduction, this point cannot be ignored.

Thank you very much for your comment. Although we significantly improve the permeability of ions in the new nanoporous Lumirror[®] film, its selectivity between the alkali metal ions is actually less than the previously discovered Hostaphan[®] films from our group. This actually shows the polymer nanopores still have a large room to compete with their nature counterpart. We add a sentence “We expect that using other polymer films or composite materials, it is possible to tune the selectivity and permeability and finally reach the demand of large-scale real-world applications such as the extraction of Li⁺ ions or the potential to compete with their natural counterparts.” in paragraph 1 on page 16.

8. On page 12, the authors suggest that “the nanopores in the PET Lumirror film in this study are nearly uniformly sized”. There is no direct evidence to back up this claim from material characterization. How uniform are the pores? What is the spacing between pores?

Assume the nanopore is distributed evenly, the average spacing between the pores is ~45 nm when the irradiation dose is $5 \times 10^{10} \text{ cm}^{-2}$. We think the pore size is uniform based on the observation that the measured transport rates are highly reproducible on different membranes. Unfortunately we could not succeed in imaging the nanopores directly with TEM or AFM. But several lines of evidence suggest that the size nuclear latent track can be quite uniform. For example, the SEM image shows the chemical etched tracks on polymer film are almost the same in size (see figure below); the TEM images on the nuclear tracks on PI membrane stained with RuO₄ and OsO₄ show bright uniform-sized spots (Adla, A. et al. Nuclear Instruments and Methods in Physics Research Section B: Beam Interactions with Materials and Atoms 185, 210–215 (2001)). To clarify this issue, we revised paragraph 1 on page 13 by adding sentences “In contrast, track-etched membranes have a very uniform pore size, but are limited in low pore density due to the stochastic pore distribution and the risk of overlapping of neighboring pores. In addition, the pore sizes of these membranes are too larger for ionic separation.”

Representative SEM image of the chemical etched tracks on polymer films (Wang, C., et al. Lab on a Chip 12, 1710 (2012))

9. The statement on page 15 that the additives in PET Lumirror are responsible for enhanced UV absorption is highly speculative. What are these additives?

For more details, please refer to our answer to the 4th remark of reviewer 2. In brief, these additives are the business secret of the company making the polymer films. We are still working on the identification of these additives, however it has been difficult because their amount could be very small. Based on our experience, we think the most important additives added in Lumirror[®] films could be some antioxidants such as Irganox, which are usual strong UV absorber, and are efficient scavengers of radicals. To clarify this issue, we rewrote paragraph 2 on page 15 discussing about the role of additives.

10. In Table S1, the hydrated radius of methylene blue ion should be 7x3 Angstrom instead of 0.7x0.3 Angstrom.

Thank you very much for your careful reading. We corrected it in the revised Supplementary Table 1.

11. The authors should provide details of the chemical structure of the polymer and force fields used in the Supplementary Information. This will help other researchers reproduce the work.

Thank you very much for your suggestion. We added the chemical structure of the polymer in the new Supplementary Figure 2a. We also added Supplementary Table 5 to show the related parameters of the ions in the CHARMM36 force fields used in our study. And we added a claim in the Material and Method section (the last paragraph on page 18) that our simulation package is available upon asking.

Reviewers' comments:

Reviewer #1 (Remarks to the Author):

Most of comments were addressed very well. I am happy to recommend to publish in NM.

One more minor comment. The authors claimed in the manuscript many times that the membrane has a high pore density. I did not find How much the pore density was and how it was measured. These details are needed.

Reviewer #2 (Remarks to the Author):

Most of the comments have been addressed well.

1. While the mechanism for ions transportation is still not clear. Even they measured the electrochemical impedance spectra (EIS). But they did not get any conclusion from them. It could get the conductivity of the ions. Then comparing the I-V measured results with EIS results.

2. It is better measure the membrane zeta potential as well as the isopoint of pH.

3. How about the surface charge density of the pores? If so close, how strong the eletrostatic interaction between the metal ions and negatively charged groups on the surface of the pores should be?

Reviewer #3 (Remarks to the Author):

The authors have satisfactorily addressed the issues raised by this reviewer. There is a concern that remains and the authors may not be able to address it in a short time. So I leave it to the editor's discretion. The concern is that the authors have not imaged the pores and so the discussion about pore size, uniformity and distribution is speculative. The BET measurements indicate a low specific surface area for PET Lumirror.

On page 3, Results section, sixth line, change 'customer-built' to 'custom-built'.

This manuscript is recommended for publication despite the concern about the pores mentioned above.

This manuscript was revised accordingly to address the remaining concerns raised by the three thoughtful reviewers. We performed the experiments and data analyses suggested by the second reviewer and present all the results in the reply. **All the requested revisions are outlined specifically below and depicted in red font in the revised manuscript.** All the other revisions (mainly related to wording changes to meet the editorial policies) are depicted in blue font.

Owing in large part to the valuable suggestions of all three reviewers and significant effort on our part to improve this paper, we are confident that this manuscript now merits publication in Nature Communications.

Summary of Specific Changes Made to the Manuscript in Response to the Reviews

To make it as easy as possible for the Editor, we pasted the reviews verbatim below. **Our response to each criticism/suggestion is depicted in green font** and the exact changes to the manuscript sections are pasted in quotations exactly as they are found in the revised manuscript, **mostly in red font.**

Reviewers' comments:

Reviewer #1 (Remarks to the Author):

Most of comments were addressed very well. I am happy to recommend to publish in NM.

One more minor comment. The authors claimed in the manuscript many times that the membrane has a high pore density. I did not find How much the pore density was and how it was measured. These details are needed.

We thank the reviewer for this valuable suggestion. Since the energy of the irradiated ions is well above the threshold to generate tracks in the PET membrane (Supplementary Figure 5b), it is well accepted that each ion generates one track. Actually, the irradiation

fluence in our experiment is always calibrated with the density measurement of the track-etched pores. As the figure shown below, the measured pore density is equal to the irradiation fluence in the range of 10^6 - 10^9 ions cm^{-2} when the track-etching method is applicable. In addition, the current in the I-V measurements is proportional to the irradiation fluence in the range of 10^9 - 5×10^{10} ions cm^{-2} (Supplementary Figure 1d), and the reproducible transport experiment indicates that the pore size is uniform (Supplementary Figure 5a). Hence it is reasonable to assume that the pore density is equal to the irradiation fluence. Since the irradiation fluence is as high as 5×10^{10} ions cm^{-2} in our samples fabricated with the track-UV method, the pore density is up to 5×10^{10} cm^{-2} , which is much greater than those made with the track-etching method. However, the expression of “high pore density” is rather arbitrary and may confuse readers, hence we explicitly show how much the pore density is when necessary and explain how it was measured in the first part of Methods. For example, we change the original sentence “we demonstrate that the high permeability is attributed to a high density of nanopores with a radius of around 0.5 nm” to “we demonstrate that the high permeability is attributed to nanopores with a radius of ~ 0.5 nm and a density up to 5×10^{10} cm^{-2} ” in the Abstract on Page 1. And we added “The track density on the membrane irradiated at low fluence was measured with the density of the pores made of the track-etching technique and it was confirmed to be consistent with the irradiation fluence” in the last paragraph on Page 12.

Figure. The measured pore density as a function of the irradiation fluence (Qi Wen. Ph.D. Thesis, Peking University, Beijing, China, 2015).

Reviewer #2 (Remarks to the Author):

Most of the comments have been addressed well.

1. While the mechanism for ions transportation is still not clear. Even they measured the electrochemical impedance spectra (EIS). But they did not get any conclusion from them.

It could get the conductivity of the ions. Then comparing the I-V measured results with EIS results.

We agree with the reviewer that EIS could provide further support for the transport mechanism. As we reported in the last reply, the resistance of different ions at high frequency in the EIS measurement is qualitatively consistent with results from the I-V measurements. To extract the conductivity of ions for EIS measurements, we need to construct an equivalent circuit model for our membrane. This turns out to be nontrivial as the EIS of our PET membrane is different from those of the typical electrochemical systems and the Nyquist plot is a little bit noisy in the low-frequency end. Nevertheless, we tried to fit the data with the equivalent circuit model consisting of a modified Randles cell with two constant phase elements (see the figure and table below). However, it is difficult to interoperate these fitting results without further well-designed control experiments. Hence we think it would be better to address this issue with more systemic studies in our future follow-up work when we have more samples.

Figure. The measured (dots) and fitted (line) Nyquist plots for the PET Lumirror[®] membranes in 0.1 M KCl (black), MgCl₂ (red) and CdCl₂ (blue). The equivalent circuit used in the fitting is shown as an inset at the left upper corner.

Table. Fitting results of the equivalent circuit.

Electrolyte solution	R _s (Ω)	R _p (Ω)	CPE-1		CPE-2	
			Y ₀ (mS cm ⁻² s ⁻ⁿ)	n	Y ₀ (μS cm ⁻² s ⁻ⁿ)	n
0.1M KCl	28.1	89.8	49.0	0.317	23.1	0.831
0.1M MgCl ₂	36.0	68.0	75.1	0.230	20.6	0.781
0.1M CdCl ₂	45.9	66.4	64.4	0.277	13.5	0.823

2. It is better to measure the membrane zeta potential as well as the isoelectric point of pH.

The determination of the isoelectric point of pH and the pH-dependence of the membrane zeta potential inside the nanopore could be a complimentary check for the pH-dependent transport measurement. The membrane zeta potential through the nanopore is commonly measured with the streaming potential measurement (measure the potential across the membrane by setting a pressure difference) (Philippe Degardin, *et al.*, Langmuir, 2005). However, it has been very difficult to perform such measurements on our PET membranes with 1-nm-diameter pores. After trial and error with several other candidate methods and equipment, we can successfully measure the pH dependence of the surface membrane zeta potential of the PET Lumirror® films with Zetasizer Nano ZS90 equipped with a surface zeta potential cell (Malvern Instruments, for more details of the measurement, see the figure caption below). Our idea is to use the surface of the membrane irradiated by energetic ions to mimic the nanopore wall of the membrane. But the pore density of the current membrane sample is too small (total pore area only accounts for less than 0.04% of the total area) and high-energy irradiation could not be scheduled till next year, we prepared PET membrane samples by irradiating with 35 keV H^+ ions (which can also cause irradiation damage on the surface) at a much higher fluence of 10^{15} ions cm^{-2} and subsequently 4-hour UV radiation. As the figure shown,

Figure. Comparison of the membrane zeta potential of the pristine (red) and irradiate (blue) PET Lumirror® films. The potential was measured at 1 mM KCl solutions with different pH values, as observed by 300 nm tracer particles DTS1235. The error bars showed the standard deviation of at least 3 independent measurements. The colored dashed lines are the fits of the data. The black dashed line shows 0, red and blue arrows show the corresponding isoelectric point of the pH of the pristine (red) and irradiate (blue) PET Lumirror® films, respectively.

the zeta membrane potential of the pristine PET membrane changes from -50 mV at pH>5 to more than 0 at pH<3, with the isoelectric point of pH being ~3.1. These results are consistent with previous measurements (Kolská, Zdeňka. *et al.*, Journal of Nano Research, 2013). Our measurement result shows that the isoelectric point of pH of the irradiated membrane shifts to a smaller value of 2.3, suggesting new charge groups are generated during ionic irradiation. It is interesting to note that the curve of the pH-dependent zeta potential is similar to the one of the pH-dependent ionic transport rates (Fig. 2c) albeit a small shift in the transition pH values, suggesting that the surface membrane zeta potential of the surrogated membrane qualitatively agrees with the zeta potential of the nanopore wall.

3. How about the surface charge density of the pores? If so close, how strong the electrostatic interaction between the metal ions and negatively charged groups on the surface of the pores should be?

It is well accepted that the surface charge density of the polymer pores made with the track-etching method is $\sim 1.0 \text{ e nm}^{-2}$ (Zuzanna S. Siwy, *Advanced Functional Materials*, 2006; T Gamble, *et al.*, *J. Phys. Chem. C*, 2014). And the polymer nanopores made with our newly developed track-UV method also show similar pH-dependent transport rates and selectivity of cations over anions (V.V. Berrzkin, *et al.*, *Colloid J. USSR*, 1991; Qi Wen, *et al.*, *Advanced Functional Materials*, 2016), indicating that these new polymer nanopores are negatively charged. And the MD simulation results are consistent with the experimental results using the polymer nanopore models with the surface charge density of $\sim 1.0 \text{ e nm}^{-2}$. We also tried to estimate the surface charge density of the pores based on the membrane zeta potential measurements. The membrane zeta potential at pH>5 in 1 mM KCl is about -70 mV, hence the average surface charge density of the membrane is calculated to be 0.06 e nm^{-2} (J. M.M.Petters, *et al.*, *Colloids and Surfaces*, 1999). This value is less than the well-accepted charge density value of the reported polymer nanopores. It is possible due to the irradiation effect difference of the 35 keV H^+ ions and the 1.4-GeV Bi ions, the concentration difference of the electrolyte solution (1 mM in the zeta potential measurement and 1 M in the transport measurement), and the environmental difference between the membrane surface and the wall of the nanopores inside the membrane. We think future systematic work could help to clarify this discrepancy, e.g., extrapolating the measurement results on the membrane irradiated with a series fluence of GeV heavy ions in a series of concentrations of the electrolyte solutions when more samples are available.

The negatively charged groups will absorb some of the metal ions through the electrostatic interaction as we demonstrated in the MD simulation. We think the electrostatic interaction is similar to the one in the formation of the electrical double layer on the charged surface.

Reviewer #3 (Remarks to the Author):

The authors have satisfactorily addressed the issues raised by this reviewer. There is a concern that remains and the authors may not be able to address it in a short time. So I leave it to the editor's discretion. The concern is that the authors have not imaged the pores and so the discussion about pore size, uniformity and distribution is speculative. The BET measurements indicate a low specific surface area for PET Lumirror.

On page 3, Results section, sixth line, change 'customer-built' to 'custom-built'.

This manuscript is recommended for publication despite the concern about the pores mentioned above.

We agree with the reviewer that it would be better if we could image the membrane to directly reveal the size and structure of the nanopores. Although the transport experiments with organic ions and different-sized PEG molecules clearly indicate that the pore radius is about 0.5 nm and this result is also consistent with the MD simulations using the polymer nanopore model, we have tried a variety of imaging methods including TEM, AFM, SEM, Helium microscopy and etc. But these traditional methods lack sufficient spatial resolution and contrast to image the polymer nanopore of 0.5 nm in radius. We are now developing two more advanced techniques: 1) improve the imaging contrast in TEM such as cutting the PET membrane to thinner sheets and staining the nanopore with dyes or heavy metal ions; 2) develop *in situ* small X-ray scattering (SAXS) to measure the pore size of the nanopores in solution under applied bias. This could be an excellent following-up work in the near future.

We have changed 'customer-built' to 'custom-built' and made some other wording changes to meet the editorial policies.

REVIEWERS' COMMENTS:

Reviewer #2 (Remarks to the Author):

All the comments have been well addressed. It is acceptable.